# Dynamics Modeling and Theoretical Study of the Two-Axis Four-Gimbal Coarse–Fine Composite UAV Electro-Optical Pod

**Cheng Shen**, **Shixun Fan \***, **Xianliang Jiang, Ruoyu Tan and Dapeng Fan**

College of Intelligence Science and Technology, National University of Defense Technology, Changsha 410073, China; shensicheng1996@sina.cn (C.S.); jxl123gfkd@163.com (X.J.); tyirorie@hotmail.com (R.T.); fdp@nudt.edu.cn (D.F.)

**\*** Correspondence: shixunfan@nudt.edu.cn; Tel.: +86-158-7428-5588

**Abstract:** In the UAV electro-optical pod of the two-axis four-gimbal, the characteristics of a coarse–fine composite structure and the complexity of dynamics modeling affect the entire system's high precision control performance. The core goal of this paper is to solve the high precision control of a two-axis four-gimbal electro-optical pod through dynamic modeling and theoretical study. In response to this problem, we used finite element analysis (FEA) and stress study of the key component to design the structure. The gimbals adopt the aerospace material 7075-t3510 aluminum alloy in order to meet the requirements of an ultralight weight of less than 1 kg. According to the Euler rigid body dynamics model, the transmission path and kinematics coupling compensation matrix between the two-axis four-gimbal structures are obtained. The coarse–fine composite self-correction drive equation in the Cartesian system is derived to solve the pre-selection and check problem of the mechatronic under high-precision control. Finally, the modeling method is substituted into the disturbance observer (DOB) disturbance suppression experiment, which can monitor and compensate for the motion coupling between gimbal structures in real time. Results show that the disturbance suppression impact of the DOB method with dynamics model is increased by up to 90% compared to PID (Proportion Integration Differentiation method) and is 25% better than the traditional DOB method.

**Keywords:** two-axis four-gimbal; electro-optical pod; dynamics modeling; coarse–fine composite

## 1. Introduction

The UAV electro-optical pod system is widely used in ship-borne, vehicular, and airborne equipment and also plays a necessary role in recent information technology equipment [1]. It can accept the region target image information, accurately identify the target motion state, and guide decision making. Previous studies in the literature [2–6] used the PIOGRAM diagram method to explain the kinematics principle of the stable mechanism and pointed out the geometric coupling problem of a two-axis two-gimbal structure. Through special bearing and motor design, previous studies in the literature [7,8] constructed a two-axis two-gimbal stable tracking platform with large field-of-view visual axis. One study [9] applied the Euler dynamics theorem to establish the equation of the visual axis stabilization structure. However, the two-axis two-gimbal structure is suitable for a stable platform with low speed and low demand for stability precision, which may cause too much error or even self-locking when working under normal conditions. In the UAV two-axis four-gimbal electro-optical pod, the characteristics of a coarse–fine composite structure and the complexity of dynamics modeling affect the high precision control performance of the system. Therefore, it is necessary to adopt new dynamics modeling and theory to study the two-axis four-gimbal coarse–fine composite electro-optical pod for use in a UAV.

A previous study [10] adopted external prestressed steel, which is applied to concrete cylinder pipes. This derivation configures the prestress of steel strands to meet the requirements of ultimate limit states, serviceability limit states, and quasi-permanent limit states, considering the tensile strength of the concrete core and the mortar coating, respectively. Another study [11] presents a simplified mathematical model for the analysis of varying compliance vibrations of a rolling bearing. The results of the parametric analysis demonstrate that, with the proper choice of the size of the internal radial clearance and external radial load, the level of the varying compliance vibrations in a rolling bearing can be theoretically reduced to zero. In the literature [12,13], an aluminum conductor steel-reinforced cable and a racing tire are modeled to study their vibrations and finite element analysis. The above modeling methods are worth being referred to. However, these methods do not study the two-axis four-gimbal electro-optical pod for use in a UAV, and there is lack of experiments on dynamics modeling of a coarse–fine composite structure platform.

Another previous study [14] analyzed the equal-acceleration model of a two-axis four-gimbal maneuvering target. Taking the equivalent sinusoidal movement and the uniform linear movement as examples, the system was simulated. The results show that the precision of the coarse–fine composite control is higher than that of single-detector control, and the two-axis four-gimbal structure is simple and suitable for engineering implementation. Reference [15] presents the magnetic field analysis for the double layer Halbach array voice coil motor. The analytical model is built by adopting Fourier analysis and proves the feasibility of the analytic method with the equivalent structure. Reference [16] is an analysis and modeling the fast steering mirror. A detailed analysis was provided to show the proposed approach and improve disturbance suppression performance with only a slight weakening of the target tracking ability. The proposed feed-forward control was effectively verified through a series of comparative simulations and experiments. Besides, the method was applied in a real ship-based project. However, this dynamic modeling and the theoretical study of these methods are applicable to medium or large platforms and devices. It is of little significance to the design of an ultralight two-axis four-gimbal coarse–fine UAV electro-optical pod.

In this paper, the dynamics modeling and theoretical study of the two-axis four-gimbal coarse–fine composite UAV electro-optical pod is deeply analyzed. In response to this problem, finite element analysis (FEA) and theoretical analysis of the stress and deflection of the key structural component are used to design the structure. According to the Euler rigid body dynamics model, the transmission path and motion coupling compensation matrix between two-axis four-gimbal are obtained, and suitable aerospace materials were used for analysis. Finally, the simulation verifies the correctness of the model.

## 2. Structure Design

As shown in Figure 1, the two-axis four-gimbal coarse–fine composite structure can be simplified to a cantilever beam. Because the integrated shafting structure requires high precision, and there are deflection errors in actual processing and manufacturing, it is necessary to check the mechanical parameters of the uniaxial structure to observe whether it meets the performance requirements of the cantilever beam.

### 2.1. Bending Internal Force and Deflection

To better clarify the simplified model, the structure of the two-axis four-gimbal electro-optical pod structure is divided into five key components for analysis. As shown in Figures 1 and 2, S1 is the spherical cover, S2 is the outer pitch gimbal that is the core component of the simplified model of the cantilever beam, S3 is the fine-stage components (think of it as the load q in the middle of S2), S4 is the voice coil motor that outputs constant torque F, S5 is the end cover that is on the left side and the fixed end of the cantilever beam. What is more, because the rotation angle between the gimbals is relatively small, the torque change is ignored, and its maximum value is taken for analysis.

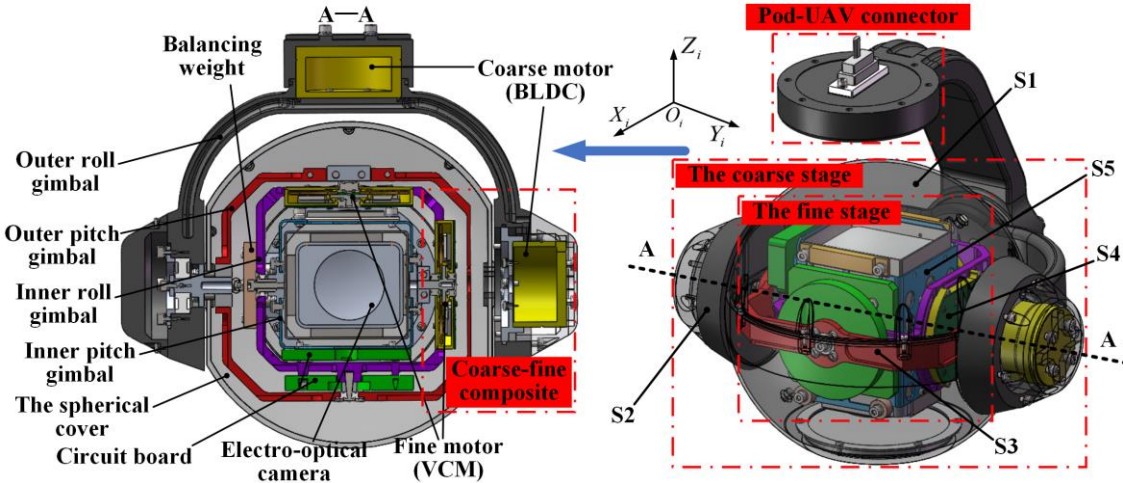

**Figure 1.** Ultralight two-axis four-gimbal electro-optical pod and coarse–fine composite system. BLDC: brushless direct current motor; VCM: voice coil motor.

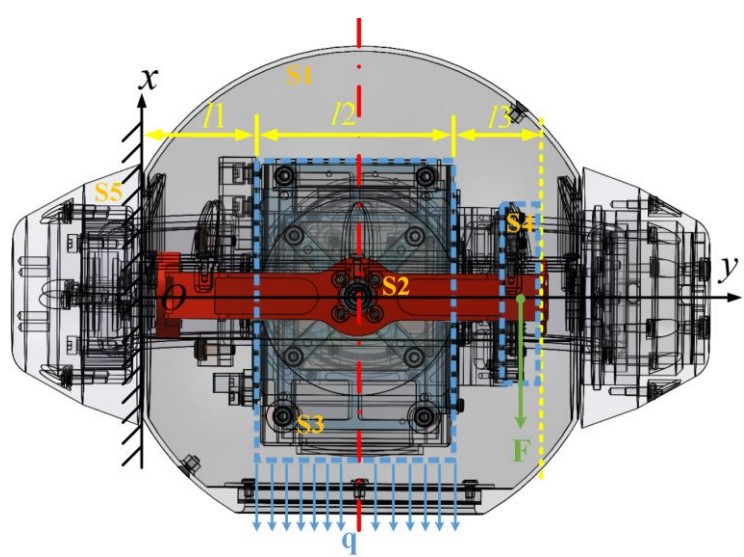

**Figure 2.** The clarifications of the model as simplified to a cantilever beam.

First, the bending internal force of plane bending under external force is analyzed. Moreover, the internal force diagram of bending moment and shear force is drawn by force analysis. In additions, the core problem is to check the deflection error of the simplified model of the cantilever beam.

In Figures 2 and 3, suppose that the connection between S2 and S5 is the origin O. Then, establish the Cartesian coordinate system Oxy. The distance of $l1$, $l2$, and $l3$ are shown in Figure 2. $l1$ is the distance between the fixed end of the left end cover and the fine-stage components. $l2$ is the length of fine-stage components. $l3$ is the distance between the fine-stage components and the voice coil motor (VCM). Span H is the sum of $l1$, $l2$, and $l3$. F is the VCM output constant torque. q is the load that is enforced by the fine-stage components S2 in the middle of the cantilever beam S2 (outer pitch gimbal).

In Figure 3, the *x*-axis is the length of the outer pitch gimbal, which is the simplified model of the cantilever beam. $F_x$-x and M-x represent the changing states of shear force $F_x$ and bending moment M at different positions, X, of in the cantilever beam. What is more, the internal force diagram of bending moment and shear force is drawn. Figure 3 shows the change of bending moment $M$ and shear force $F_x$ with *x*.

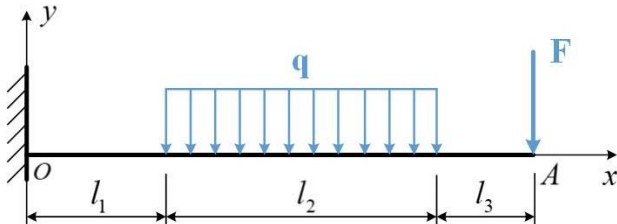

**Figure 3.** Force analysis of coarse–fine composite structure.

Because the deformation of the cantilever beam is very small, the change of the beam's span length after deformation is ignored. Since the example reports a cantilever beam, at the O-point (the fixed end), the bending moment is max. Figure 4a,b presents the changing states of the shear force $F_x$ and bending moment M. Moreover, the material of the beam works within the elastic range of the beam, so the deflection and angle of the beam are linear with the load acting on the beam. Using static equilibrium analysis of material mechanics, the deflection of cantilever beam is calculated by the superposition principle. Because of its complicated force, it is divided into three force forms to solve the equations, respectively, which are finally superimposed together to obtain the deflection curve equation of the coarse–fine composite structure.

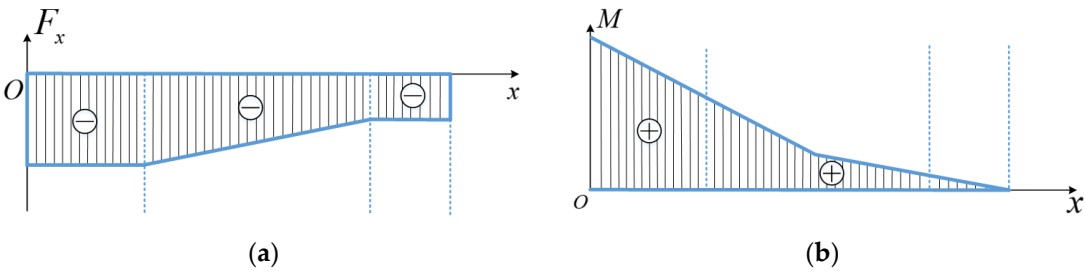

(**a**)　　　　　　　　　　　　　　　　　　　　　(**b**)

**Figure 4.** The internal force diagram. (**a**) $F_x - x$; (**b**) $M - x$.

In Figure 5, due its complexity, the force of the system is divided into three force forms in order to use the superposition principle to solve the equation. What is more, the rigid displacement of the free end A of the cantilever beam is selective analysis. The key is to decompose the load q in the middle into two loads q starting at the origin A. The first is down, the second is up.

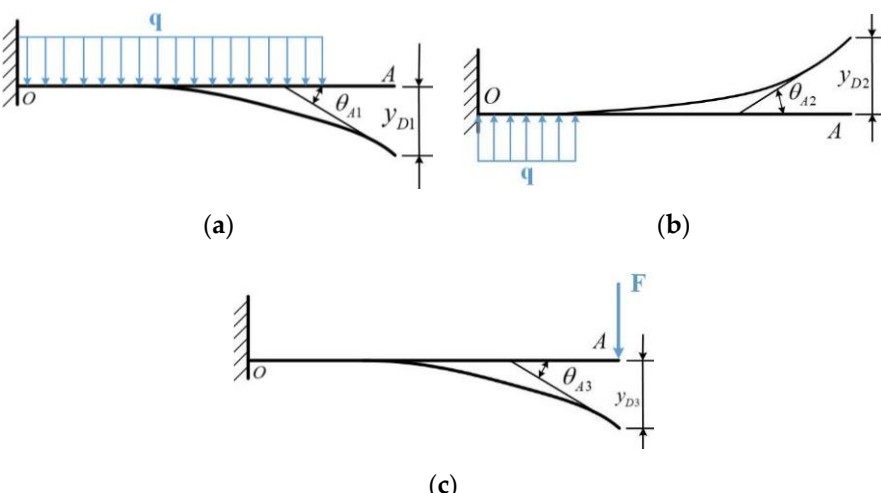

(**a**)　　　　　　　　　　　　　　　　　　　　　(**b**)

(**c**)

**Figure 5.** Superposition principle method static equilibrium analysis. (**a**) Form 1; (**b**) Form 2; (**c**) Form 3.

As shown in Table 1, the structure distance data obtained after Solidworks software simulation is analyzed. The deflection curve equation of each superposition diagram is as follows:

$$y_{D1} = y_1 + \theta_{A1} \cdot l_3 = \frac{q(l_1 + l_2)^4}{8EI} + \frac{q(l_1 + l_2)^3}{6EI} \cdot l_3 = \frac{3q(l_1 + l_2)^4 + 4q(l_1 + l_2)^3 l_3}{24EI}, \tag{1}$$

$$y_{D2} = y_2 + \theta_{A2} \cdot (l_2 + l_3) = -\frac{q(l_1)^4}{8EI} - \frac{q(l_1)^3}{6EI} \cdot (l_2 + l_3) = -\frac{3q(l_1)^4 + 4q(l_1)^3(l_2 + l_3)}{24EI}, \tag{2}$$

$$y_{D3} = y_3 = \frac{F(l_1 + l_2 + l_3)^3}{3EI}, \tag{3}$$

**Table 1.** Structure distance data obtained by Solidworks software.

| Length | Value/mm |
|--------|----------|
| $l_1$ | 40 |
| $l_2$ | 43 |
| $l_3$ | 33 |
| Span H | 116 |

By superimposing Equations (1)–(3), we can obtain

$$y_D = y_{D1} + y_{D2} + y_{D3} = \frac{3q(l_1+l_2)^4+4q(l_1+l_2)^3 l_3}{24EI} - \frac{3q(l_1)^4+4q(l_1)^3(l_2+l_3)}{24EI} + \frac{F(l_1+l_2+l_3)^3}{3EI}, \tag{4}$$

were E = the elastic modulus of the material, $N/mm^2$; I = the cross-sectional area of the material, $mm^2$; and q = standard values for distributed loads, $kN/m$. As shown in Table 2, after Solidworks software simulation, the free end force q and the average distributed load F were obtained as follows:

**Table 2.** Structure free and force and average distributed load data by Solidworks software.

| Parameter | Value |
|-----------|-------|
| F | 0.00022 kN |
| q | 0.0085 kN/m |

It is made of aluminum alloy nonferrous metal with excellent comprehensive performance and its brand name is 7075-t3510. According to the data, the elastic modulus of 7075 aluminum alloy is $E = 71.7Gpa$. The coarse–fine composite structure adopted the calculation method of moment of inertia of circular section. All known parameters are substituted into the equation of the deflection curve derived from the superposition.

$$y_D = y_{D1} + y_{D2} + y_{D3} \approx 1.67 \times 10^{-6} mm, \tag{5}$$

The coarse–fine stage composite structure is the overall plane bending of the main structure. Therefore, the allowable deflection is less than H/1500. The calculated result is

$$\frac{H}{1500} = \frac{0.116}{1500} \approx 0.77 \times 10^{-4} mm > 1.67 \times 10^{-6} mm. \tag{6}$$

To sum up, the deflection of the coarse–fine composite structure is checked to meet the specified deviation requirement. According to the internal force diagram of bending moment and shear force, the bending internal force of the cantilever beam under the action of external force is within the normal range.

### 2.2. Finite Element Analysis

In the Figure 6, finite element analysis (FEA) was carried out for the force of the key structural component, and the objects were meshed and solved by FEA. We then analyzed whether the stress, strain, and displacement parameters met the requirements.

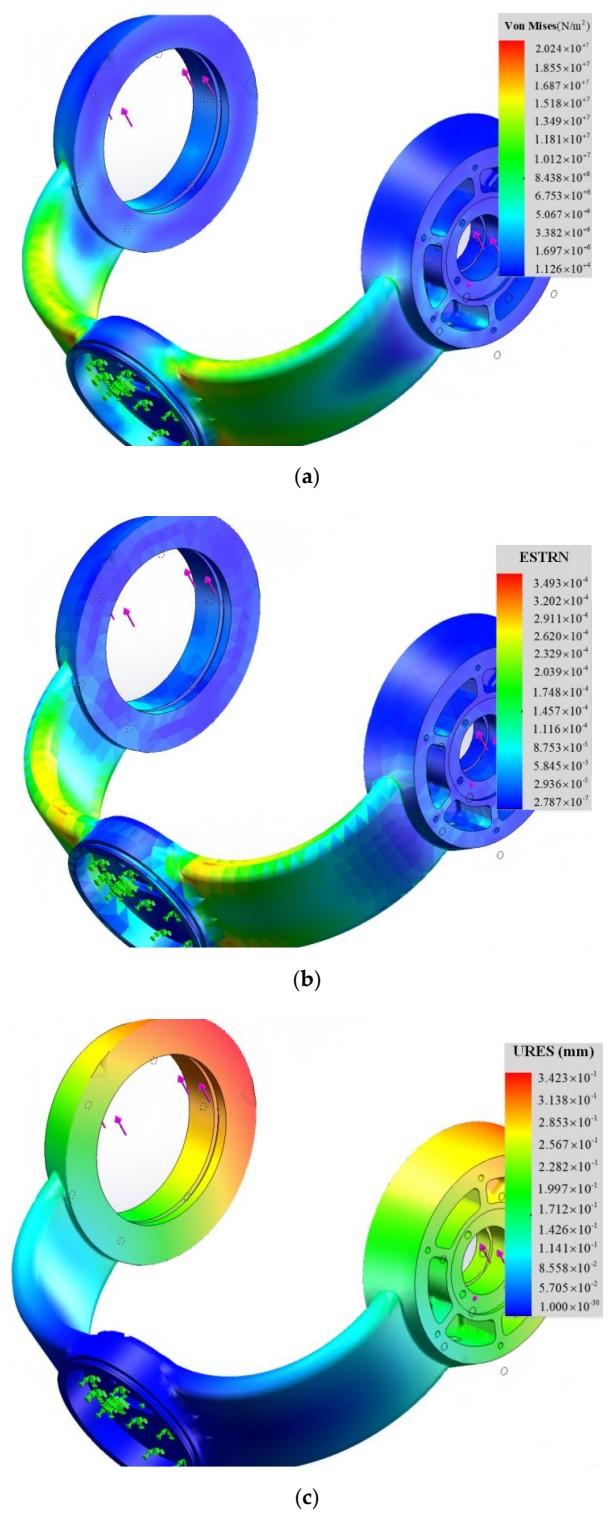

**Figure 6.** Finite Element Analysis (FEA) of key structure component (outer roll gimbal). (**a**) Stress analysis; (**b**) strain analysis; (**c**) displacement analysis.

It can be seen from the grading on the right of Figure 6a–c and Table 3 that the more red the structure is, the more dangerous it is. The increase of stress, strain, and displacement in the drilling position is large and relatively concentrated, but it still meets the needs for the normal working of the structure within the safety range. The results of finite element analysis still prove that it can meet the requirements of operation, and the outer roll gimbal is safe and reliable as a whole.

**Table 3.** Finite element analysis data obtained by Solidworks software.

| Parameter | Value | |
|---|---|---|
| | Max | Min |
| Stress | $2.02 \times 10^7 N/m^2$ | $1.13 \times 10^4 N/m^2$ |
| Strain | $3.49 \times 10^{-4}$ | $2.79 \times 10^{-7}$ |
| Displacement | $3.42 \times 10^{-1} mm$ | $1.00 \times 10^{-30} mm$ |
| The deformation ratio | 49.3719 | |

### 2.3. Design of Limit Structure of Rotation Angle

According to the Euler transformation of the fixed-point rotation of a rigid body, the Euler angle has no limit. However, in the coarse–fine composite structure of the two-axis four-gimbal electro-optical pod, the rotation angle of each gimbal is limited due to the external dimension, load weight, and the center of mass imbalance of the gimbal.

As shown in Figure 7a, in order to ensure the normal operation of the UAV's electro-optical pod in a safe range, a limit stopper is used to limit the rotation angle of each gimbal structure. A balancing weight is used to allocate the overall mass to prevent the occurrence of center of mass imbalance. As shown in Figure 7a,b, the rotation angle of inner pitch gimbal limiting stopper is +7°~−7° (a total of 14°), and the rotation angle of inner roll gimbal limiting stopper is +12°~−12° (a total of 28°). In this angle range, the operation of the UAV's electro-optical pod is normal and safe.

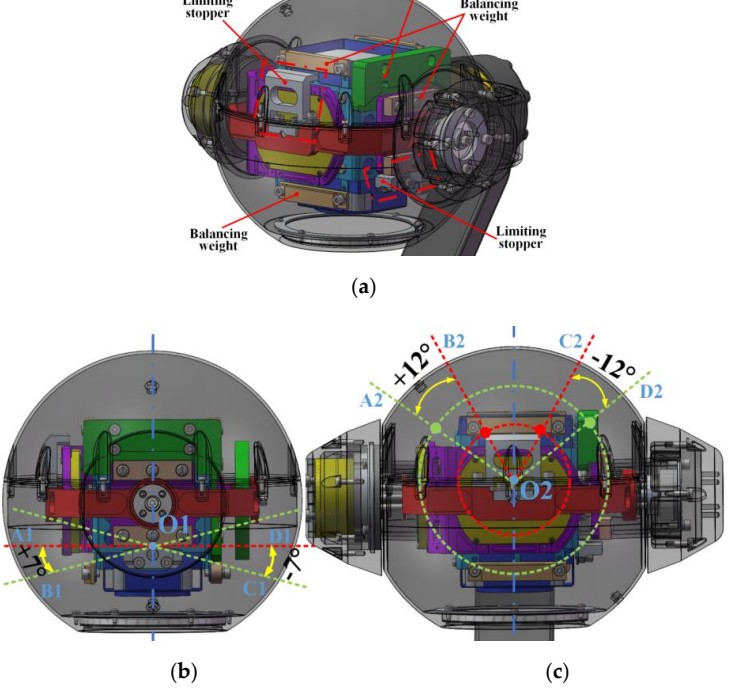

**Figure 7.** The diagram of rotation angle of each gimbal of the electro-optical pod. (**a**) The electro-optical stopper; (**b**) the rotation angle of the inner pitch gimbal limiting stopper; (**c**) the rotation angle of the inner roll gimbal limiting stopper.

## 3. Dynamics Modeling

### 3.1. Coarse–Fine Composite Analysis

As shown in Figure 1, the structure of the two-axis four-gimbal electro-optical pod is more complicated. It is therefore an effective solution to study the coarse–fine composite structure first. The working principle of the coarse–fine composite structure involves the definition of multiple coordinate systems, which are respectively explained as follows.

A    Inertial coordinate system ($\{i\}$, $O_iX_iY_iZ_i$)
B    UAV coordinate system ($\{d\}$, $O_dX_dY_dZ_d$)
C    Coarse motor stator coordinate system ($\{u\}$, $O_uX_uY_uZ_u$)
D    Fine motor rotor coordinate system ($\{g\}$, $O_gX_gY_gZ_g$)

The coarse motor is fixedly connected with the guide rail through the threaded connection, without considering the damping effect between the structures. There is geometric eccentricity $e_r$ in the shafting structure of the coarse–fine stage mechatronic system, which will cause coaxiality error and affect the high precision control performance of the electro-optical pod. As shown in Figure 8a, the geometric eccentricity of the shafting structure is caused by force deformation, uneven cutting force, and chip formation of the cutting edge.

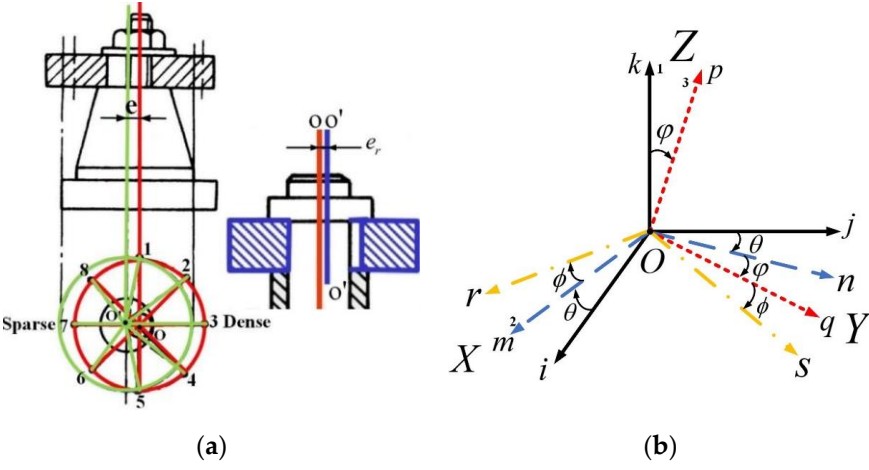

(**a**)                    (**b**)

**Figure 8.** (**a**) Geometric eccentricity error of coarse–fine composite structure; (**b**) Euler transformation diagram for fixed-point rotation.

In Figure 8b, according to the transformation matrix of fixed-point rotation in the Cartesian coordinate system, the Euler transformation [17] is analyzed as shown in Figure 8b, and the Euler angle is $\theta, \phi, \varphi$. The first step is to rotate the $\theta$ angle about the $k$ axis, so that the $i$ axis rotates to the $m$ position and the $j$ axis rotates to the $n$ position; Cartesian coordinate system $Oijk \rightarrow Omnk$. The second step is to rotate the $\phi$ angle about the $m$ axis, so that the $n$ axis rotates to the $q$ position and the $k$ axis rotates to the $p$ position; Cartesian coordinate system $Omnk \rightarrow Omqp$. The third step is to rotate the $\varphi$ angle about the $p$ axis, so that the $m$ axis rotates to the $r$ position and the $q$ axis rotates to the $s$ position; Cartesian coordinate system $Omqp \rightarrow Orsp$. Finally, the Euler transformation of fixed-point rotation is completed.

In the Cartesian coordinate system, after the system $\{u\}$ rotates the $\theta_1$ angle around the $X_i$ axis, the system $\{i\}$ is used as the reference system to observe the position of the system $\{u\}$. Then, the system $\{u\}$

rotates the $\theta_2$ angle around $Z_i$ axis. Euler transformation of coarse–fine composite structure can be calculated as the rotation transformation matrix, denoted as

$$E^{X\theta} = A^\theta = \begin{pmatrix} 1 & 0 & 0 \\ 0 & \cos\theta_1 & -\sin\theta_1 \\ 0 & \sin\theta_1 & \cos\theta_1 \end{pmatrix}, \tag{7}$$

$$E^{Z\theta} = A^\theta = \begin{pmatrix} \cos\theta_2 & -\sin\theta_2 & 0 \\ \sin\theta_2 & \cos\theta_2 & 0 \\ 0 & 0 & 1 \end{pmatrix}. \tag{8}$$

According to Euler transformation law of rigid body fixed point rotation, it can be obtained from Equations (7) and (8) that

$$E = E^{k\theta} \cdot E^{m\phi} \cdot E^{p\varphi} = \begin{pmatrix} \cos\theta & -\sin\theta & 0 \\ \sin\theta & \cos\theta & 0 \\ 0 & 0 & 1 \end{pmatrix} \cdot \begin{pmatrix} 1 & 0 & 0 \\ 0 & \cos\phi & -\sin\phi \\ 0 & \sin\phi & \cos\phi \end{pmatrix} \cdot \begin{pmatrix} \cos\varphi & -\sin\varphi & 0 \\ \sin\varphi & \cos\varphi & 0 \\ 0 & 0 & 1 \end{pmatrix}. \tag{9}$$

Since the stator of the coarse motor is connected with the guide rail by thread, there is no fixed-point rotation for the system {$u$} against the system {$i$}. Only the installation error of rotation along the $Y$-axis exists. The kinematic coupling equation shows that

$$\omega_u = \begin{pmatrix} \omega_{ux} \\ \omega_{uy} \\ \omega_{uz} \end{pmatrix} = \begin{pmatrix} \cos\theta_u & 0 & \sin\theta_u \\ 0 & 1 & 0 \\ -\sin\theta_u & 0 & \cos\theta_u \end{pmatrix} \begin{pmatrix} \omega_{ix} \\ \omega_{iy} \\ \omega_{iz} \end{pmatrix} + \begin{pmatrix} 0 \\ \dot\theta_u \\ 0 \end{pmatrix}. \tag{10}$$

Due to the Euler transformation law of rigid body fixed point rotation, the kinematics coupling equations of the system {$u$} against the system {$v$} and the system {$v$} against the system {$g$} are

$$\omega_v = \begin{pmatrix} \omega_{vx} \\ \omega_{vy} \\ \omega_{vz} \end{pmatrix} = E^{k\theta v} E^{m\phi v} E^{p\varphi v} \begin{pmatrix} \omega_{ux} \\ \omega_{uy} \\ \omega_{uz} \end{pmatrix} + \begin{pmatrix} \dot\theta_{vx} \\ \dot\theta_{vy} \\ \dot\theta_{vz} \end{pmatrix}, \tag{11}$$

$$\omega_g = \begin{pmatrix} \omega_{gx} \\ \omega_{gy} \\ \omega_{gz} \end{pmatrix} = E^{k\theta g} E^{m\phi g} E^{p\varphi g} \begin{pmatrix} \omega_{vx} \\ \omega_{vy} \\ \omega_{vz} \end{pmatrix} + \begin{pmatrix} \dot\theta_{gx} \\ \dot\theta_{gy} \\ \dot\theta_{gz} \end{pmatrix}, \tag{12}$$

The symbols used in the equation and Figure 9 are defined as follows: $\dot\theta_u$ = the angular velocity vector of the coarse motor stator relative to the inertial coordinate system; $\dot\theta_{vx}, \dot\theta_{vy}, \dot\theta_{vz}$ = the angular velocity vector of the coarse motor rotor relative to the coarse motor stator coordinate system; $\dot\theta_{gx}, \dot\theta_{gy}, \dot\theta_{gz}$ = the angular velocity vector of the fine motor rotor relative to the coarse motor rotor coordinate system; $\omega_u, \omega_v, \omega_g$ = the angular velocity and its components on the coordinate axis; and LOS = the line of sight.

In Figure 9, the external environment disturbance is included. In order to simplify the analysis process of the coarse–fine stage visual axis stabilization, this paper mainly discusses the conduction path and characteristics of UAV motion to the mechatronic system. Therefore, the disturbance input of the external environment is analyzed as an inertial coordinate system, and the whole process of motion coupling of the coarse–fine mechatronic system is obtained through the transformation of Cartesian coordinate along the system {$u$}, system {$v$}, and system {$g$}. The Euler angle $\theta, \phi, \varphi$ is determined separately in order to determine the relationship between the angular velocities at each stage and the inertial space.

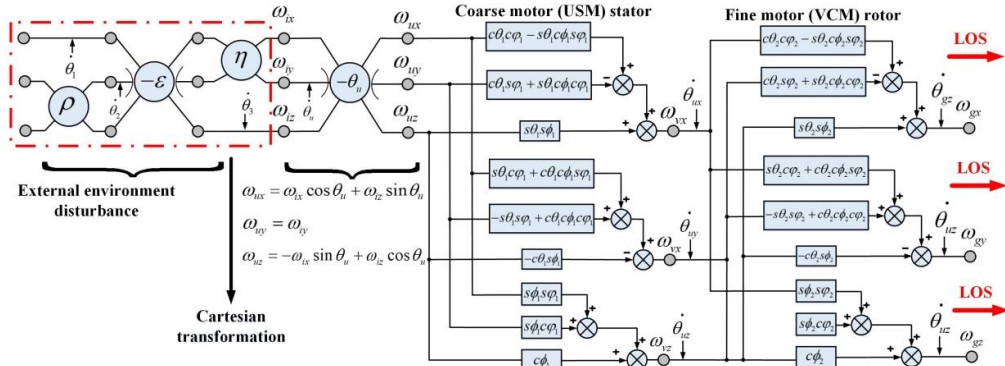

**Figure 9.** The coarse–fine composite structure of electro-optical pod kinematic coupling Piogram.

## 3.2. Two-Axis Four-Gimbal Structure

Based on the analysis of the transmission path and kinematics coupling compensation matrix of the coarse–fine composite structure, the dynamics modeling and theoretical study of the ultralight two-axis four-gimbal electro-optical pod are studied. The working principle of the two-axis four-gimbal electro-optical pod involves the definition of multiple coordinate systems, which are respectively explained as shown in Figure 10.

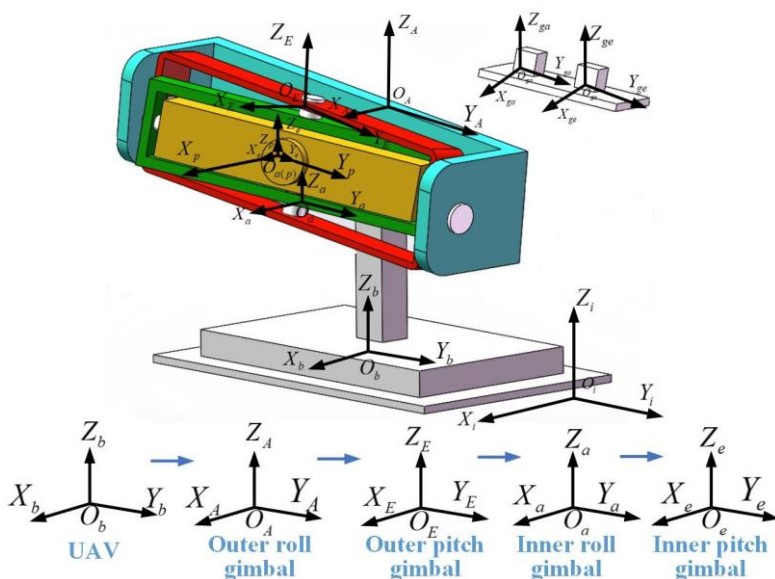

**Figure 10.** Simplified Cartesian coordinate system of the two-axis four-gimbal electro-optical pod.

The inner gimbal rotates in a small range, and the outer gimbal follows the macro-field control of the inner gimbal in a large range. At the same time, the feedback error of the outer gimbal is compensated by the inner gimbal so that the inner gimbal can offset the disturbance of rolling and pitching. Finally, the two-axis four-gimbal electro-optical pod maintains the stability of the visual axis to achieve high-precision coarse–fine composite control.

**A. Direct Connection Stabilization**

The gyroscope is sensitive to the angular velocity of the inner pitch system {e} and the inner roll system {a} relative to the inertial system {i}. Therefore, make $\omega_{Ye} = \omega_{Ze} = 0$, and then the structure can keep the visual axis of the detector stabilization.

$$
\begin{pmatrix} \dot{\theta}_e \\ \dot{\theta}_a \end{pmatrix} = \left\{ \begin{array}{c} \omega_{gyro\_Y} \\ \omega_{gyro\_X} \end{array} \right. = \begin{pmatrix} \sin\theta_a \cos\theta_E \\ \cot\theta_e \cos\theta_a \cos\theta_E + \sin\theta_E \end{pmatrix} \omega_{ZA} \\
+ \begin{pmatrix} -\cos\theta_a & -\sin\theta_E \sin\theta_a \\ \cot\theta_e \sin\theta_a & -\cot\theta_e \cos\theta_a \sin\theta_E - \cos\theta_E \end{pmatrix} \begin{pmatrix} \omega_{YE} \\ \omega_{XA} \end{pmatrix} ,
\tag{13}
$$

According to Equation (13), the structure can keep the visual axis of the stabilization.

**B. Indirect Connection Stabilization**

The angular velocity of gyroscope sensitive the outer pitch axis system {E} and the outer roll axis system {a} relative to the inertial system {i} is $\omega_{gyro\_X} = \omega_{XA}, \omega_{gyro\_Y} = \omega_{YE}$.

$$
\begin{pmatrix} \dot{\theta}_e \\ \dot{\theta}_a \end{pmatrix} = \begin{pmatrix} 1 & 0 \\ 0 & -\sec\theta_e \end{pmatrix} \begin{pmatrix} \omega_{Ye} \\ \omega_{Ze} \end{pmatrix} + \begin{pmatrix} \sin\theta_a \cos\theta_E \\ \cot\theta_e \cos\theta_a \cos\theta_E + \sin\theta_E \end{pmatrix} \omega_{ZA} \\
+ \begin{pmatrix} -\cos\theta_a & -\sin\theta_E \sin\theta_a \\ \cot\theta_e \sin\theta_a & -\cot\theta_e \cos\theta_a \sin\theta_E - \cos\theta_E \end{pmatrix} \begin{pmatrix} \omega_{gyro\_Y} \\ \omega_{gyro\_X} \end{pmatrix} ,
\tag{14}
$$

According to Equation (14), the structure can keep the visual axis of the stabilization.

Assuming that, in the case of sensitive motion of pitch and roll gyroscopes, their sensitivity values are $\omega_{gyro\_Y}$, $\omega_{gyro\_Z}$, respectively, $\dot{\theta}_a$ represents the angular velocity vector of the inner roll gimbal relative to the outer pitch gimbal, and $\dot{\theta}_e$ represents the angular velocity vector of the inner roll gimbal relative to the inner roll gimbal; $\omega_A$, $\omega_E$, $\omega_a$, $\omega_e$, respectively, represent the angular velocity of the two-axis four-gimbal structure and the components of its three coordinate axes.

According to the Euler dynamical theorem and Coriolis rotation law,

$$
\frac{dH}{dt} = \frac{\partial H}{\partial t} + \omega \times H,
\tag{15}
$$

where $H = \begin{pmatrix} H_X & H_Y & H_Z \end{pmatrix}^T$ = moment of momentum; $\frac{dH}{dt}$ = absolute derivatives (rate of change) of vector H; $\frac{\partial H}{\partial t} = \frac{dH_X}{dt}i + \frac{dH_Y}{dt}j + \frac{dH_Z}{dt}k$ = relative derivatives of vector H; and $i, j, k$ = unit vectors of coordinate axis of the body reference system, respectively.

According to the moment of momentum theorem,

$$
\frac{dH}{dt} = M,
\tag{16}
$$

where $M = \begin{pmatrix} M_X & M_Y & M_Z \end{pmatrix}^T$ = the external addition torque vector of the rigid body. Under the assumption that the all three axes are principal axes of inertia, the following equation can be established:

$$
\begin{cases} I_X \dot{\omega}_X + (I_Z - I_Y)\omega_Y \omega_Z = M_X \\ I_Y \dot{\omega}_Y + (I_X - I_Z)\omega_X \omega_Z = M_Y \\ I_Z \dot{\omega}_Z + (I_Y - I_X)\omega_X \omega_Y = M_Z \end{cases} ,
\tag{17}
$$

where $I_X, I_Y, I_Z$ = the moment of inertia of the rigid body around the coordinate axis of the follower reference system.

The moment of momentum theorem is applicable to the calculation of larger angular velocity. However, the electro-optical pod of two-axis four-gimbal structure is compact and ultralight, with a mass less than 1 kg and a stabilization precision is 20 μrad. When the design is carried out in

combination with the actual situation, the values of some parameters are ignored. Based on the space dynamics [18], the coarse–fine composite self-correction drive equation are derived.

### A. Inner Pitch Gimbal

$$M_{Ye} \approx (x_e \times \dot{v}_b)m_e + [x_e \times (\dot{\omega}_a \times y_e)]m_e + [x_e \times (\dot{\omega}_E \times y_a)]m_e + [x_e \times (\dot{\omega}_A \times y_E)]m_e + I_{Ye}\dot{\omega}_e + (\omega_e \times I_{Ye}\omega_e), \quad (18)$$

### B. Inner Roll Gimbal

$$\begin{aligned}
M_{Xa} &\approx (x_a \times \dot{v}_b)m_a + [x_a \times (\dot{\omega}_E \times y_a)]m_a + [x_a \times (\dot{\omega}_A \times y_E)]m_a + (y_e \times \dot{v}_b)m_e \\
&+ [y_e \times (\dot{\omega}_a \times y_e)]m_e + [y_e \times (\dot{\omega}_e \times y_e)]m_e + I_{Xa}\dot{\omega}_a + [\omega_a \times (I_{Xa}\omega_a)] + M_{Ye}
\end{aligned}, \quad (19)$$

### C. Outer Pitch Gimbal

$$\begin{aligned}
M_{YE} &\approx (x_E \times \dot{v}_b)m_E + [x_E \times (\dot{\omega}_A \times y_E)]m_E + (y_a \times \dot{v}_b)m_a + [y_a \times (\dot{\omega}_E \times y_a)]m_a \\
&+ [y_a \times (\dot{\omega}_a \times y_a)]m_a + (y_a \times \dot{v}_b)m_e + [y_a \times (\dot{\omega}_E \times y_e)]m_e + [y_a \times (\dot{\omega}_a \times y_e)]m_e \\
&+ [y_a \times (\dot{\omega}_e \times y_e)]m_e + I_{YE}\dot{\omega}_E + [\omega_E \times (I_{YE}\omega_E)] + M_{Xa} + M_{Ye}
\end{aligned}, \quad (20)$$

### D. Outer Roll Gimbal

$$\begin{aligned}
M_{XA} &\approx (x_A \times \dot{v}_b)m_A + (y_E \times \dot{v}_b)m_E + [y_E \times (\dot{\omega}_A \times y_E)]m_E + [y_E \times (\dot{\omega}_E \times y_E)]m_E \\
&+ (y_E \times \dot{v}_b)m_a + [y_E \times (\dot{\omega}_A \times y_a)]m_a + [y_E \times (\dot{\omega}_E \times y_a)]m_a + [y_E \times (\dot{\omega}_a \times y_a)]m_a \\
&+ (y_E \times \dot{v}_b)m_e + [y_E \times (\dot{\omega}_A \times y_e)]m_e + [y_E \times (\dot{\omega}_E \times y_e)]m_e + [y_E \times (\dot{\omega}_a \times y_e)]m_e \\
&+ [y_E \times (\dot{\omega}_e \times y_e)]m_e + I_{XA}\dot{\omega}_A + [\omega_A \times (I_{XA}\omega_A)] + M_{YE} + M_{Xa} + M_{Ye}
\end{aligned}, \quad (21)$$

where $\omega_i$ = angular velocity of inner pitch{e}, inner roll{a}, outer pitch{e}, outer roll{a} relative to inertial gimbal system{i}; $v_b$ = the speed of the UAV gimbal {b} relative to the inertial gimbal; $m_i$ = the quality of inner pitch, inner roll, outer pitch and outer roll gimbal; $I_{..} = I_{Ye}, I_{Xa}, I_{YE}, I_{XA}$ is the rotational inertia of the inner pitch, the inner roll, the outer pitch and the outer roll gimbal along their respective rotation axis; $x_i$ = the vector distance from the origin of four gimbal coordinate systems and UAV coordinate systems to their respective centroids is designated as the inner pitch $x_e$, inner roll $x_E$, outer pitch $x_a$, outer roll $x_A$, and UAV $x_b$; $y_i$ = the vector displacement between the rotation axis of the inner pitch gimbal and the inner roll gimbal is $y_e$, the vector displacement between the rotation axis of the inner roll and the outer pitch gimbal is $y_a$, the vector displacement between the rotation axis of the outer pitch and the outer roll gimbal is $y_E$; and $M_{..}$ = the torque of the four gimbals relative to the rotation axis in the inertial coordinate system is the output torque of the four motors.

In order to further study the Euler rigid body dynamics model mechanism of the ultralight two-axis four-gimbal electro-optical pod, Figure 11 is drawn. In Figure 11, the torques due to gimbal kinematics and those due to geometrical coupling have been combined. The key problem is to ensure the high precision control of the structure visual axis.

When both $\theta_e = 0$ and $\theta_a = 0$, the inner gimbal angle is zero. The disturbing moment of the two axes is minimized. Through the following performance of the outer gimbal, the mutual perpendicularity between the inner gimbals can be guaranteed, so as to eliminate the geometric constraint coupling brought by the outer gimbal to the visual axis and realize the interference isolation, proving once again that the system can decouple two stabilization channels from the perspective of kinematics. The coupling interference of geometric constraints can be eliminated, and the control precision can be improved.

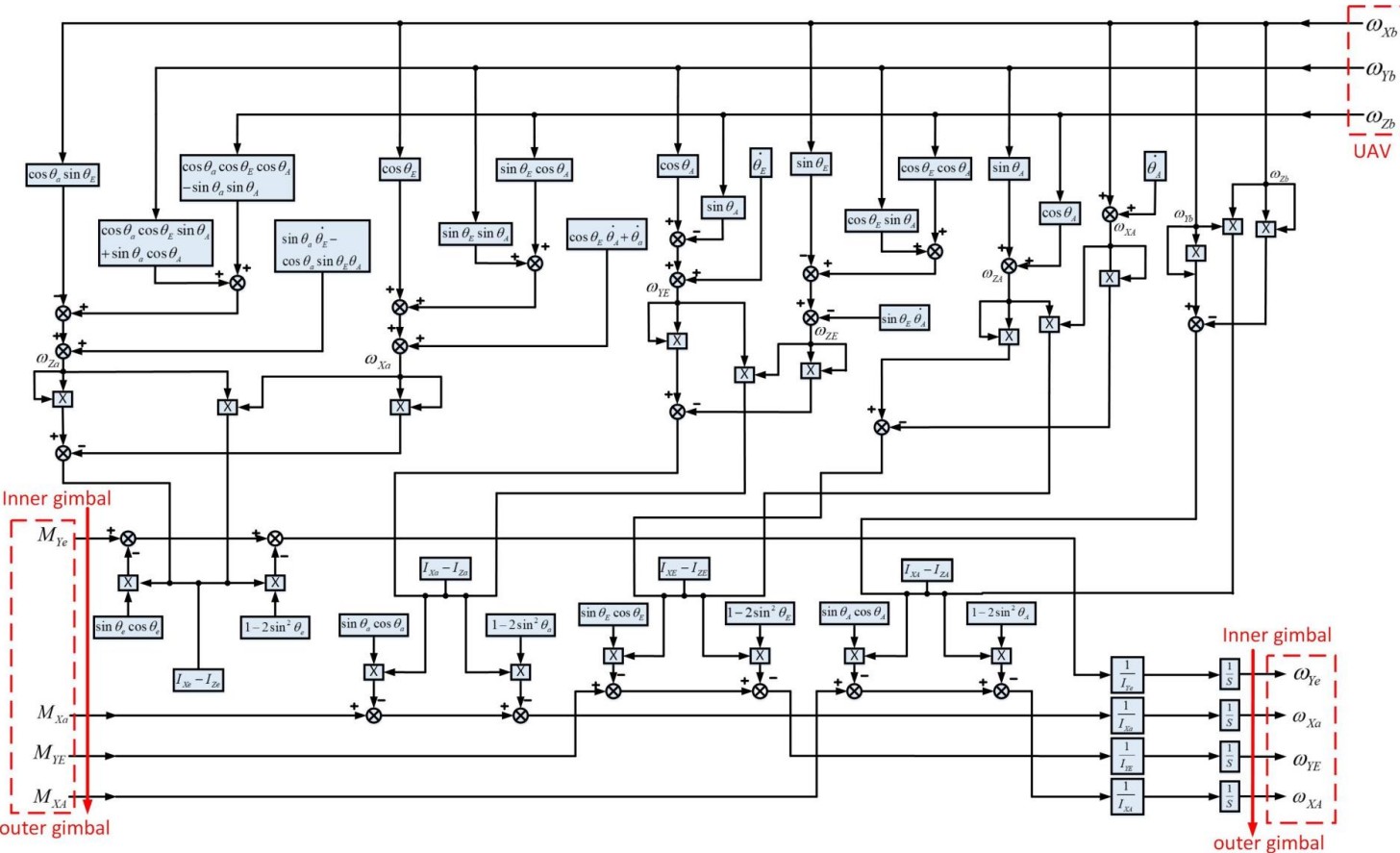

**Figure 11.** The ultralight two-axis four-gimbal electro-optical pod torque relationships.

### 3.3. Comparison Validation

Because the motor adopts a direct drive way, it does not consider slip failure. The impact caused by friction is small, so it is assumed that the transmission efficiency of the final stage of the pod is $\eta = 99\%$. The parameters of gimbal rotation angular velocity, UAV maneuvering acceleration, and gimbal angular velocity are

$$\dot{\omega}_{Ye} = \dot{\omega}_{Xa} = \dot{\omega}_{YE} = \dot{\omega}_{XA} = \varepsilon = 120°/s^2 \approx 2rad/s^2 \approx 0.318r/s^2, \tag{22}$$

$$\dot{v}_b = a \leq 5g \approx 50m/s^2, \tag{23}$$

$$\omega_{Ye} = \omega_{Xa} = \omega_{YE} = \omega_{XA} = \omega_{max} = 60°/s \approx 1rad/s \approx 0.159r/s, \tag{24}$$

As shown in the coarse–fine composite drive self-correction equation, the cross-product term value is small, and the included angle is small as it approaches zero infinitely and is greater than zero. According to the trig function, if $\theta \to 0, \cos\theta \to 1$. What is more, $a \times b = |a| \cdot |b| \cdot \cos\langle a, b\rangle$. Therefore, the term $\cos\langle a, b\rangle$ can be ignored as the constant 1.

As shown in Table 4, the moment of inertia and mass data of gimbals at all stages when the electro-optical pod rotates at 0° are presented. The moment of inertia and the distance between the center of mass and the origin are analyzed when the electro-optical pod rotates at different angles. We then calculate the coarse–fine composite forecast torque (Equations (18)–(21)), full payload, and equivalent dynamics load calculates torque (Equations (22)–(24)) of the electro-optical pod structure. Our results are shown in Figure 12.

**Table 4.** Rotational inertia when the initial rotation angle is 0° and quality simulation data.

| Gimbal | Rotational Inertia (Including Load/kg·m$^2$) | | | Mass (Including Load/kg) |
| --- | --- | --- | --- | --- |
| | X | Y | Z | |
| Inner pitch e | $0.57 \times 10^{-3}$ | $0.58 \times 10^{-3}$ | $0.48 \times 10^{-3}$ | 0.471 |
| Inner roll a | $0.89 \times 10^{-3}$ | $0.84 \times 10^{-3}$ | $0.93 \times 10^{-3}$ | 0.681 |
| Outer pitch E | $0.99 \times 10^{-3}$ | $0.94 \times 10^{-3}$ | $0.11 \times 10^{-2}$ | 0.740 |
| Outer roll A | $0.26 \times 10^{-1}$ | $0.88 \times 10^{-2}$ | $0.27 \times 10^{-1}$ | 1.925 |

It can be observed from Figure 12 that the difference of the coarse–fine composite forecast torque, full payload, and equivalent dynamics load calculates the torque. This is due to the influence of friction, wind load, and conductor's interference torque. The load of the inner pitch gimbal is less at the center and the influence of disturbance is minimal, so the difference with the real value is not large. The gimbal is extended one level outward, the bearing load increase, the shape is more irregular, the circuit board leads are complex, and other factors cause the error to increase within a certain range of the true value.

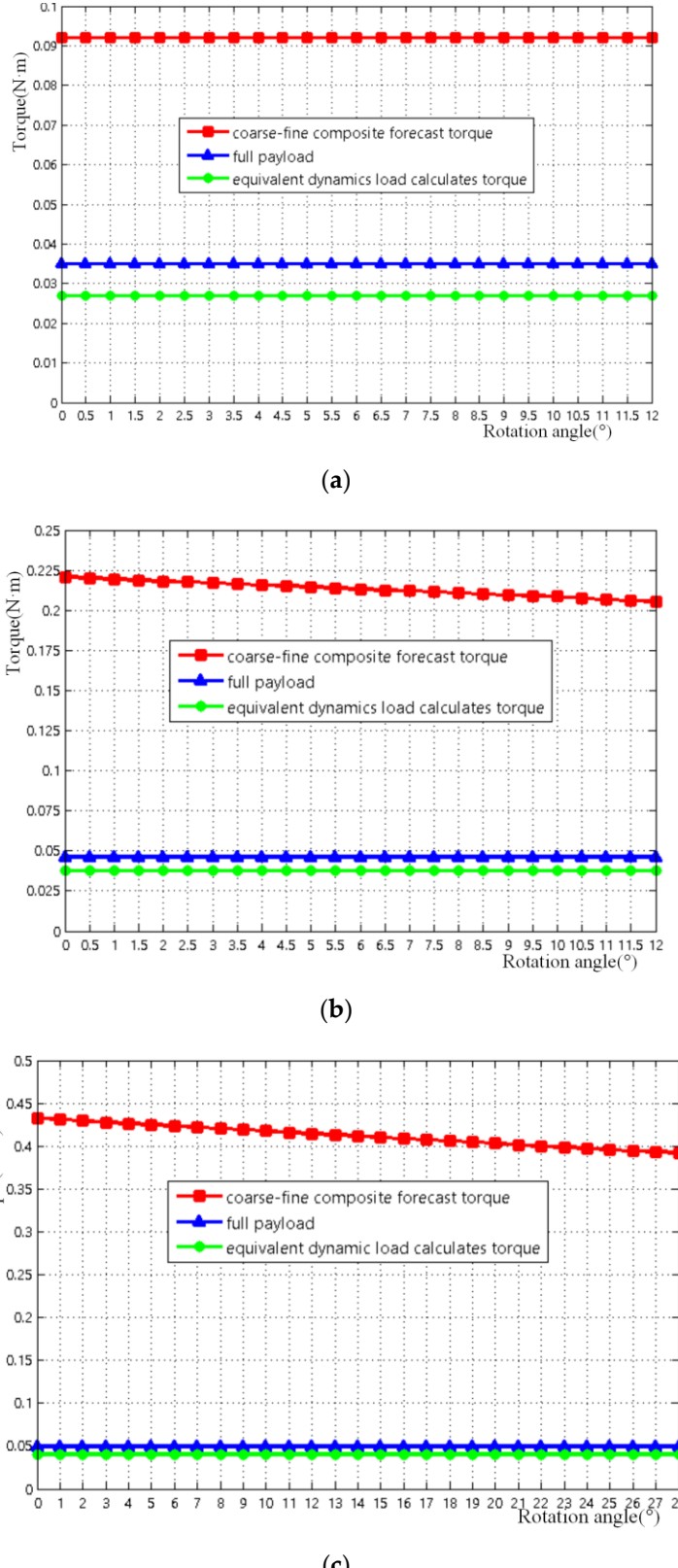

(**a**)

(**b**)

(**c**)

**Figure 12.** *Cont*.

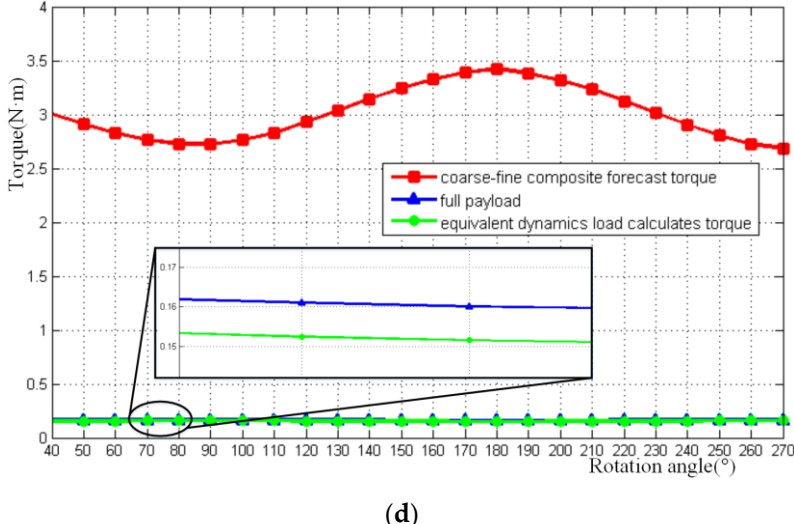

**(d)**

**Figure 12.** The ultralight two-axis four-gimbal electro-optical pod torque comparison validation. (**a**) Inner roll gimbal; (**b**) inner pitch gimbal; (**c**) outer roll gimbal; (**d**) outer pitch gimbal.

## 4. Experiment

In view of the problem that the coupling effect between two-axes four-gimbal seriously affects the stability precision, it is necessary to combine the coupling relationship of rotational inertia of each gimbal axis for disturbance suppression analysis. Because of the effectiveness of the interference observer (DOB) in suppressing external interference [19,20], in this paper, an interference observer suitable for the ultralight two-axis four-gimbal electro-optical pod is studied.

As shown in Figure 13, the control object is set as the ultralight two-axis four-gimbal system, and the minimum phase system under ideal state is adopted. The nominal inverse model of the controlled object is $Js + B$. Based on the kinematic coupling analysis and modeling, a DOB disturbance observer is used to study self-correcting disturbance suppression. The traditional DOB controller is improved to a time-varying DOB controller with rotational inertia. At the same time, the results of the moment of inertia analysis after the modeling mentioned in this paper are substituted into the nominal inverse model $J_n$ of each gimbal control loop. Realize the real-time change of $J_n$ following the change of gimbal angle $\theta$. The control loop of the outer roll gimbal A was given sinusoidal interference as an example, and a Matlab Simulink simulation comparison experiment was carried out.

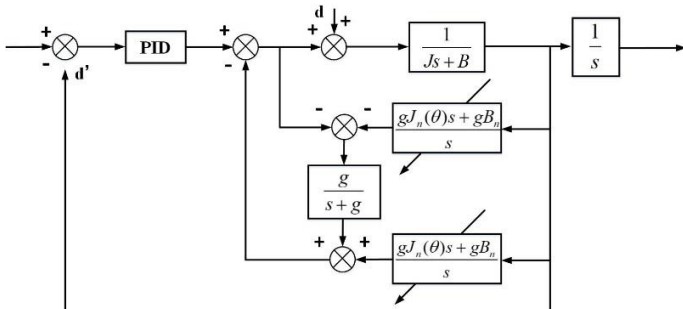

**Figure 13.** Disturbance suppression output diagram of outer roll gimbal A.

The rotational inertia of the rotating axis changes in real time. First, the parameters of the traditional PID controller are set as follows: $K_p = 20, K_i = 6$. As can be seen from Table 4, the initial parameters of rotational inertia are set as $J = J_{XA} = 0.176 \times 10^{-1} \text{kg} \cdot \text{m}^2$. The parameters of the disturbance observer and its low-pass filter are set as follows: $B = 0.002, g = 200, B_n = 0.002$. From the

motion coupling and modeling analysis, it can be known that the coupling rotational inertia on the outer roll gimbal A is

$$
\begin{aligned}
J =\ & (\cos\theta_E \cos^2\theta_e - \sin\theta_E \cos\theta_a \sin\theta_e \cos\theta_e)J_{Xe} + \\
& (\cos\theta_E \sin^2\theta_e - \sin\theta_E \sin\theta_a \cos\theta_a \cos^2\theta_e)J_{Ze} + \cos\theta_E J_{XE} + J_{XA}
\end{aligned}
\tag{25}
$$

Figures 14 and 15 verify the optimality of the velocity loop's traditional PID control, the traditional DOB disturbance suppression control, the improved DOB self-correcting disturbance suppression control, and the low-pass filter parameter selection. Set the system input amplitude to 0. The input amplitude of sine wave disturbance is 10 rad/s, and the frequency is 8 Hz. As shown in Figure 14, in order to facilitate the observation of the experimental results, the output value of the improved DOB was taken as negative gain output Scope which was distinguished from the other three waveforms. Further observation of the experimental results shows that

$$
\Delta X < \Delta Y,
\tag{26}
$$

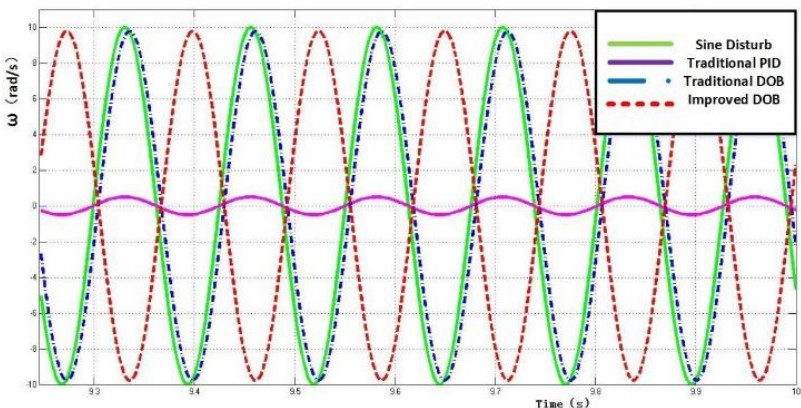

**Figure 14.** Disturbance suppression output diagram of outer roll gimbal A.

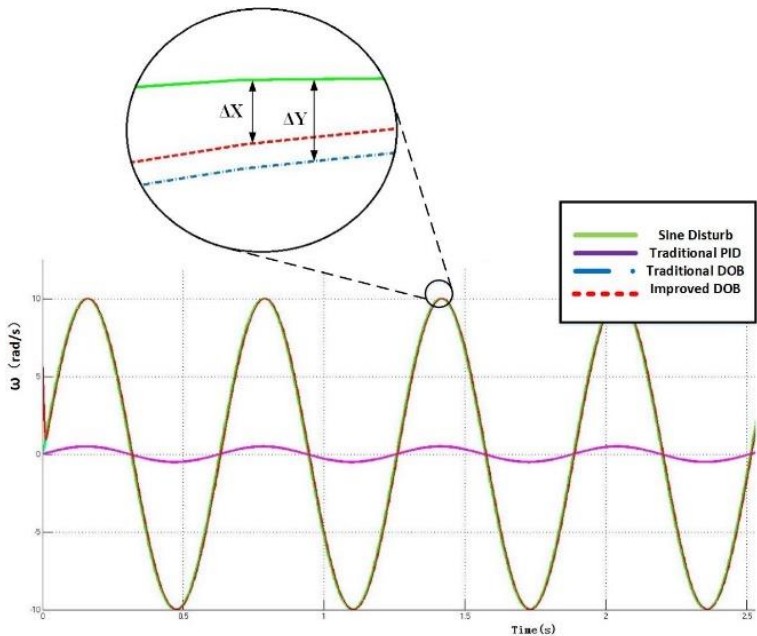

**Figure 15.** Comparison diagram of disturbance suppression output of outer roll gimbal A.

The results show that the disturbance suppression impact of DOB method with dynamics model is increased by up to 90% better than PID. As shown in Figure 13, this is defined as

$$e = d - d', \tag{27}$$

As shown in Figure 16, by comparing the two figures, it can be known that the estimated deviation of traditional DOB disturbance suppression is $e = 0.51$ and the estimated deviation of the improved DOB disturbance suppression is $e = 0.43$. The results show that the disturbance suppression impact of DOB method with dynamics model is increased by up to 25% compared to the traditional DOB.

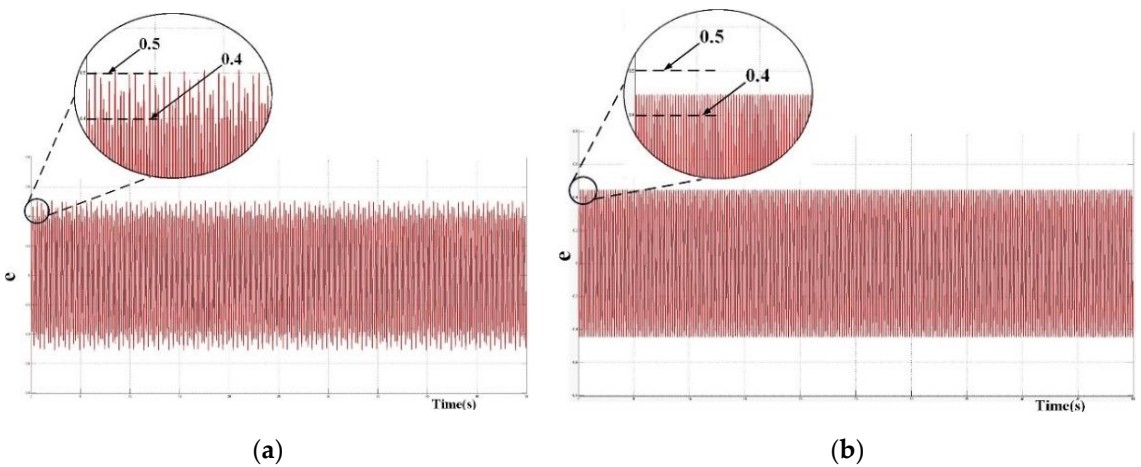

| (a) | (b) |

**Figure 16.** Comparison diagram of disturbance suppression difference output of outer roll gimbal A. (**a**) The output deviation of traditional DOB disturbance suppression; (**b**) the output deviation of improved DOB disturbance suppression.

## 5. Conclusions

This paper represents an in-depth study on the dynamics modeling and theoretical study of the two-axis four-gimbal coarse–fine composite electro-optical pod. Our conclusions are as follows.

A   In the UAV electro-optical pod of the two-axis four-gimbal, the characteristics of the coarse–fine composite structure and the complexity of dynamics modeling affect the entire system's high-precision control performance. The core goal of this paper is solve the high precision control of two-axis four-gimbal electro-optical pod through dynamic modeling and theoretical study. FEA and theoretical analysis of the stress and deflection of the key structure component was used to design the structure. The gimbal structure adopts 7075-t3510 aluminum alloy, which is an aerospace material that meets the requirements of an ultralight electro-optical pod weighing less than 1 kg.

B   According to the Euler rigid body dynamics model, the transmission path and kinematics coupling compensation matrix for the two-axis four-gimbal are obtained. The coarse–fine composite drive correction equation of the inner-outer gimbals is derived to solve the pre-selection and check problem of the coarse–fine motors under high-precision control.

C   The modeling method is substituted into the DOB disturbance suppression experiment, which can monitor and compensate for the motion coupling between gimbal structures in real time. Our results show that the disturbance suppression impact of the DOB method with dynamics model is up to 90% better than PID and 25% better than traditional DOB.

D   This manuscript is based on the dynamics modeling and theoretical study of the two-axis four-gimbal coarse–fine composite UAV electro-optical pod. This manuscript is valuable for all researchers interested in the coarse–fine composite, two-axis four-gimbal structures, and ultralight electro-optical pods.

**Author Contributions:** C.S. designed the Euler rigid dynamics model and high precision macro–micro composite DOB control algorithm and carried out experimental research on the effect of the modeling and algorithm. D.F. and S.F. guided the research and proposed the ideas and revisions of the paper. X.J. provide help with control and simulation. R.T. provide structure model. C.S. and S.F. revised the paper. All authors have read and agreed to the published version of the manuscript.

**Funding:** This research received no external funding.

**Acknowledgments:** The author would like to thank all the teachers and colleagues who provided inspirations and equipment in the experiment. The author would like to thank all the anonymous reviewers for their meticulous comments and helpful suggestions.

**Conflicts of Interest:** The authors declare no conflict of interest.

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
