# Peer review of "Dynamics Modeling and Theoretical Study of the Two-Axis Four-Gimbal Coarse–Fine Composite UAV Electro-Optical Pod"

_applsci, doi:10.3390/app10061923_

Round 1

Reviewer 1 Report

For Author:

The paper is focused on the dynamics modeling and theoretical study of the two-axis four-gimbal coarse-fine composite electro-optical pod. In particular the work is divided in 3 main section and in each of them a specific problem is discussed. In the first section a Finite element analysis and theoretical analysis on the main component of the system is provided. Then, according to Euler rigid body dynamics model, the transmission path and kinematics coupling compensation matrix between two-axis four-gimbal are obtained. Finally, the provided method is applied in for an experimental case in order to demonstrate the capability to monitor and compensate the motion coupling between gimbal structures in real time.

The achieved results are very interesting, even if the authors could improve the way they are presented.

The entire work should be improved from an explanation point of view. Sometime the discussion is not too clear and some doubts remain in the reader.

The authors should emphasize the innovative aspects more, trying not to neglect the synthesis capabilities.

  • Abstract

Comments: Please review the entire paragraph especially from a grammar point of view and try to better underline the goals of the work.

  • Line 78

The author write:  As shown in figure 1, the two-axis four-gimbal coarse-fine composite structure can be analyzed by means of material mechanics cantilever beam.

Comments:  it is not clear why a reader should understand from figure 1 that the cantilever beam theory can be applied. Please provide more clarifications about the simplified model.

  • Line 82 – Section 2.1

Comments:  It is no clear which part of the 3d model, the schematic model should represent. In table 1, the authors reports some measures, but it is not possible to understand the real meaning of them (in the real model).

  • Line 83

The author write:  Analyze the bending internal force of plane bending under the action of external force.

Comments:  it seems a title… improve the sentence.

  • Line 95

Comments: the caption of figure 3 is not clear. Do F – x and M – x mean F(x) and M(x)? therefore the shear force and the moment along the x-axis?

  • Line 90

The author write:

Comments:  the diagram in figure 3 reports the Fx (x), taking into account the figure 2 Fx should be the normal force in the beam. The authors state that the diagram reports the shear force Fs, but in the figure there is Fx. The diagram, obviously, shows the shear force… so please remove the ambiguity.

  • Figure 3

Comments:  The moment diagram is not correct. Since the example reports a cantilever beam, at the O-point (the fixed end) the reaction moment is max, while the authors report a value equal to zero.

  • Figure 5

Comments:  the reported legend are not clear. It is not impossible to read the values in the legend. Increase the font

  • Line 146 – Table 3

The author write:  The increase of stress, strain and displacement in thedrilling position is large and relatively concentrated, but it still meets the needs of normal working of the structure within the safety range

Comments:  the maximum deflection, reported in the table is equal to 3.42 E-01 mm, but previously the authors state that the allowable deformation is H/1500 = 1.67 E-06 mm, Please provide a clarification.

  • Conclusion

Comments:  the conclusion are very short and schematic. In this way the good level of obtained results is not emphasized enough. The authors could improve this section.

Author Response

Please see the attachment. Thank you for reviewing my manuscript!

Reviewer 2 Report

Though it is an interesting topic to work on and present your data. You have developed good models which can be beneficial for understanding structural properties of UAV. Fine tuning of structure of sentences and language is must and so is presentation style for smooth flow of your paper. 

1) In your paper few things should be defined upfront such as

i) Detail and definition of coarse-fine composite material should be in introduction 

ii) ultra-light in UAV what is the range of weight. 

2) You have defined cross sectional area mm^4. Area is mm^2 if its an error in typing it should be fixed but if its an error in model/theoretical calculation than than needs to be addressed. Unless you have other reasons to believe unit of area is mm^4 which should also be clarified then.

3) Your FEA analysis somewhat define the boundary but not necessarily defines the maximum limit and what will be STress/Strain /Displacement for initiation of failed Gimble. 

4) Also limits of Euler Angles should be defined such that failure is completely understood for claiming safe operation of the UAV. This would be helpful in understanding maximum rotation before failure along various axis.

5) In moment of momentum theorem you mention mass below 1 Kg, does that mean this model has limit of UAV has the limit of 1 Kg. What happens when mass is higher than 1 Kg. Is the study still applicable. Is the model/simulation scalable to varying weight of Gimble. 

The limits to failure should be defined for understanding the theoretical aspect of the study and applicability of model 

Author Response

(The authors gave the same response as above.)
